# Combined and Singular Effects of Ethanolic Extract of Persian Shallot (*Allium hirtifolium* Boiss) and Synbiotic Biomin^®^IMBO on Growth Performance, Serum- and Mucus-Immune Parameters and Antioxidant Defense in Zebrafish (*Danio rerio*)

**DOI:** 10.3390/ani11102995

**Published:** 2021-10-19

**Authors:** Hamed Ghafarifarsani, Seyed Hossein Hoseinifar, Maedeh Talebi, Morteza Yousefi, Hien Van Doan, Rudabeh Rufchaei, Marina Paolucci

**Affiliations:** 1Department of Fisheries, Faculty of Natural Resources, Urmia University, Urmia 5756151818, Iran; hamed_ghafari@alumni.ut.ac.ir; 2Department of Fisheries, Faculty of Fisheries and Environmental Sciences, Gorgan University of Agricultural Sciences and Natural Resources, Gorgan 4918943464, Iran; hoseinifar@gau.ac.ir; 3Fishery Group, Department of Natural Resources, Islamic Azad University (Lahijan Branch), Lahijan 1477893855, Iran; maede.talebi65@gmail.com; 4Department of Veterinary Medicine, Peoples’ Friendship University of Russia (RUDN University), 6 Miklukho-Maklaya St., 117198 Moscow, Russia; myousefi81@gmail.com; 5Department of Animal and Aquatic Sciences, Faculty of Agriculture, Chiang Mai University, Chiang Mai 50200, Thailand; 6Science and Technology Research Institute, Chiang Mai University, 239 Huay Keaw Rd., Chiang Mai 50200, Thailand; 7Inland Water Aquaculture Research Centre, Iranian Fisheries Science Research Institute (IFSIR), Agricultural Research Education and Extension Organization (ARREO), Bandar-e Anzali 4314166976, Iran; roofchaie@gmail.com; 8Department of Sciences and Technologies, University of Sannio, 82100 Benevento, Italy; paolucci@unisannio.it

**Keywords:** herbal medicine, synbiotic, growth, immunity, zebrafish

## Abstract

**Simple Summary:**

The present study investigated the effect of combined and singular ethanolic extract of Persian shallot and synbiotic Bomin^®^IMBO in zebrafish. The aim of this study is to measure a range of parameters consisting of growth performance, serum and mucus immune parameters, and antioxidant defense. The results indicated that the measured parameters have a positive effect and hence we can suggest administration of these additives in zebrafish culture.

**Abstract:**

This study was carried out to evaluate combined and singular effects of ethanolic extract of Persian shallot (*Allium hirtifolium* Boiss) and synbiotic Biomin^®^IMBO on growth performance, innate immune responses, and antioxidant defense in zebrafish (*Danio rerio*). Fish with initial weight of 151.90 ± 0.31 mg were allocated in 21 10-L glass aquariums. The experimental groups were as follows: T1, control (without any supplementation); T2, 1% synbiotic; T3, 3% synbiotic; T4, 1% Persian shallot (as a medical plant); T5, 3% Persian shallot; T6, 1% Persian shallot and 1% synbiotic; T7, 3% Persian shallot and 3% synbiotic. At the end of the experiment (60 days), all treatments significantly showed higher final weight (FW), weight gain (WG), WG (%), and specific growth rate (SGR) compared with the fish fed on control diet. Furthermore, both synbiotic Biomin^®^IMBO and Persian shallot significantly improved intestine immune parameters including lysozyme, alternative complement hemolytic activity (ACH50), total immunoglobulin (total Ig), and myeloperoxidase (MPO) of zebrafish compared to fish fed on control diet (*p* < 0.05). Also, in all experimental groups, hepatic catalase (CAT), superoxide dismutase (SOD), glutathione peroxidase (GPx), and glutathione reductase (GR) activities significantly increased compared to the control group. Whereas, the highest MDA level was observed in the control group compared to the treatments (*p* < 0.05). Moreover, skin mucus immune parameters of zebrafish have been noticeably improved with synbiotic Biomin^®^IMBO and Persian shallot compared to fish fed on the control diet (*p* < 0.05). The results indicate that synbiotic or Persian shallot supplemented diet could enhance the general health status of the zebrafish.

## 1. Introduction

Aquaculture is one of the main food-producing sectors experiencing rapid expansion in recent decades due to the high protein demand arising from increasing population growth [1]. The increased occurrence of diseases and low survival of fish due to intensified culture conditions are significant constraints to sustainable aquaculture production [2]. Immunostimulants present a promising approach to prevent and/or control diseases in aquaculture [1,3]. In recent years, a lot of research has been done on food compounds and supplements that play a role in improving existing health and nutritional efficiency. Various factors can affect fish production efficiency, but reducing mortality or reducing pathogens are important points that should always be considered. Nutritionists believe that increasing the efficiency of aquatic production will depend on the formulation of the diet and its production method, which depends on factors such as energy, nutrients, protein, fat, vitamins, minerals, digestibility, nature of compounds, price, and availability [4]. Traditionally, there has been an inevitable need to use antibiotics or chemical drugs to cope with pathogens, reduce stress, and increase the resistance of fish against diseases [5]. More recently, the study of phytochemicals as fish immunostimulants has been of great interest worldwide. Thus, the potential of plant extracts to enhance the growth and immunity of fish has been widely studied [6,7,8].

The application of probiotics, prebiotics, and synbiotics is one of the economic friendly alternative efforts to prevent disease in fish [9,10,11]. Probiotic is a living microorganism that can provide beneficial effects to the host, improve the balance of microbes in the gastrointestinal tract, feed efficiency, and environmental quality [11,12]. Prebiotic is a food ingredient that cannot be digested by the host, but it can increase the growth, nutrient utilization, and health status of the host by inducing the growth of beneficial microorganisms in the gastrointestinal tract [13]. The use of the combination of the two has resulted in emerging the idea of synbiotics [14]. Since then, several studies have been conducted to evaluate the effects of different synbiotics on various fish species [15,16,17].

Synbiotic (Biomin^®^IMBO) is a novel product containing (DSM530) *Enterococcus faecium* IMB52 and prebiotic Fructo-oligosaccharide to compose a symbiotic relationship or synergism, and it has been assured to improve the growth, health, and immunity status of various aquaculture species [18]. It promotes the implantation of live microbes in the gastrointestinal tract and prolongs their surveillance, stimulates the metabolism of beneficial bacteria in the gut, and enhances the secretion of digestive enzymes and nutrients absorption, improving fish growth [19,20]. Recently, many reports have been published about the beneficial effects of Biomin^®^IMBO in the diets of some commercial fish species [21]. By acting on beneficial bacteria in the gut, synbiotics increase the volume of beneficial bacteria in the gut, and ultimately, by increasing the digestibility of some beneficial compounds, they will also affect the composition of the body [22].

Sigh et al. [23] concluded that the results indicate that dietary combination of *Bacillus circulans* PB7 (BCPB7) and fructoligosaccharide (FOS) can be considered an effective synbiotic formula against low pH stress in culture practices of *L. rohita* juveniles. Changizi et al. [24] found that 1.5 g/kg diet of synbiotic administered to green terror (*Andinoacara rivulatus*) was effective on growth performance and feed efficiency. Furthermore, the results of Gheshlaghi et al. [25] showed beneficial effects of Biomin^®^IMBO on metabolism, growth, and survival of banded cichlid (*Heros severus*). Nekoubin et al. [26] found that zebrafish larvae fed the synbiotic significantly increased final weight, specific growth rate (SGR), and food conversion efficiency (FCE) in comparison with the control group.

Plant extracts have been utilized in aquaculture due to their cost-effectiveness, eco-friendliness, and potentially having no side effects [27]. On the other hand, they are known to improve growth performance, stimulate the immune system, and enhance the antioxidant capacity of fish owing to their bioactive compounds such as alkaloids, terpenoids, saponins, and flavonoid components [28]. Persian shallot or Mooseer (*Allium hirtifolium* Boiss) grows wildly as blackish, paper-like tunics in highlands in western areas of Iran and other Asian countries. The plant is one of the most important medicinal and edible plants in Iran [29], which has usually been used as a spice and flavoring agent in traditional and Asian cuisines. Some studies suggested that *A. hirtifolium* consists of 9-hexadecenoic acid, 11,14-eicosadienoic acid, and n-hexadecanoic acid, and its hydromethanolic extract exhibits strong antibacterial activities [30]. In addition, the phenolic compounds of *A. hirtifolium* extract are reported to have from moderate to good antioxidant [31] and wound healing activity [32].

Medicinal plants have been shown to have positive effects on growth performance [33,34], innate immune responses [35,36], survival against bacterial infection [37,38], repair of cutaneous lesions [39], skin mucus immunity [40], delay the microbial, lipid, and protein spoilage [41], antioxidative and antimicrobial activities, sensory scores [42], and fish preservation [43,44].

It is believed that aquatic production efficiency depends upon the formulation and method of food production [45], that is why food quality and supplements can stimulate the immune system and eventually leads to an increase in fish growth [46].

According to above, the present study was conducted to evaluate the singular or combined effects of *A. hirtifolium* Boiss extract and the synbiotic Biomin^®^IMBO on growth performance, antioxidant parameters, and innate immune responses of zebrafish (*D. rerio*).

## 2. Materials and Methods

### 2.1. Formulation and Preparation and Extraction Procedure

Plant materials were procured as fresh as possible and commercial product of synbiotic Biomin^®^IMBO, hereafter referred to as synbiotic, was purchased from the Etouk Farda Feed Additives Co. (Tehran, Iran). All chemicals used in this experiment were of analytical grade and purchased from local suppliers.

Fresh Persian shallot plants were purchased from a local market (Shahrekord, Iran). The bulbs were carefully removed, washed, chopped into pieces, and air-dried for 7 days. Dried materials were thoroughly crushed and mixed with 80% ethanol at a ratio of 1:3 (*w/v*). The mixture was intermittently shaken for three days to allow proper extraction. The mixture was then filtered using 0.45 µm filter paper, centrifuged (3000 rpm, 5 min, 4 °C) to separate the alcoholic extract from solid substances. The obtained extract was concentrated in a rotary evaporator, freeze dried, and stored at −20 °C until use.

### 2.2. Preparation of Diets

Different amounts of synbiotic and/or Persian shallot were added to the basal diet according to a previously described method [47]. A commercial diet (Aller Aqua, Allervej 130, Christiansfeld, Denmark, Co. with no additives) was used as a basal diet. In order to ensure the quality of the experimental diets, they were made every 20 days. At each stage, according to the growth of fish, the appropriate size of the commercial diet was used to prepare experimental diets. The basic diet common to all groups contained 51.1% crude protein, 13.6% total lipid, 10.2% ash, and 3.5% fiber. Different experimental diets were prepared by adding synbiotic and/or ethanolic extract of Persian shallot as follows: T1, control (without any supplementation); T2, 1% synbiotic; T3, 3% synbiotic; T4, 1% Persian shallot; T5, 3% Persian shallot; T6, 1% Persian shallot and 1% synbiotic; T7, 3% Persian shallot and 3% synbiotic.

### 2.3. Feeding Trial

A total of 420 adult zebrafish (*D. rerio)* were supplied from a local ornamental fish farm and transferred to our laboratory (Karaj, Iran). Fish were acclimatized in 100-L glass aquarium for 10 days, during which the fish were fed the control diet at the rate of 2% of biomass three times a day. Then, fish with the initial weight of 151.90 ± 0.31 mg were allocated in 21 10-L glass aquariums (7 treatments with 3 replicates) and fed with experimental diets (2% of biomass) three times a day for 60 days. Each aquarium was siphoned daily and 70% of water was replaced with fresh dechlorinated tap water. During the experiment, water physicochemical parameters including temperature, pH, dissolved oxygen, NO_2_, NO_3_, and NH_4_ were measured with a portable multi-meter, HQ40d (Loveland, CO, USA) and a Wagtech digital photometer 7100 (Berkshire, UK) as 25.4 ± 1.2 °C, 7.1 ± 1.46, 7.56 ± 0.49 ppm, and ammonia, nitrite, and nitrate (<0.1 ppm), respectively. 

### 2.4. Sampling

At the end of the trial, nine specimens were randomly selected from each aquarium and anesthetized with clove powder (0.5 g/L) [48]. After anesthesia, the fish was immediately placed in polyethylene bags containing 5 mL of 50 mM NaCl (Sigma, Steinheim, Germany) and gently rubbed into the bags for 2 min to collect the maximum amount of mucus [48]. The obtained mucus was then centrifuged, and the supernatant was collected. To obtain samples for immune and antioxidant parameters, the fish was dissected, and the liver and intestine were obtained. All samples were immediately frozen and kept at −80 °C until use [49].

### 2.5. Growth Performance

The total weight of fish in each tank was evaluated collectively at the end of the 60 day-feeding period. The following equations were used to calculate growth parameters [50]:Weight gain (WG; mg) = final weight − initial weight(1)
Specific growth rate (SGR, % day^−1^) = (Ln. final body weight − Ln. initial body weight) × 100/days(2)
Feed conversion ratio (FCR) = feed intake/weight gain(3)
Survival rate (SR) (%) = (final numbers/initial numbers) × 100 (4)

### 2.6. Analysis of Intestine and Skin Mucus Immunological and Liver Antioxidant Parameters

Determination of intestine and skin mucus lysozyme activity was performed using the method described by Esteban et al. [51]. Intestine alternative complement hemolytic activity (ACH50) was determined based on the procedures described by Yano [52]. Total immunoglobulin (total Ig) level of intestine and mucus was measured following the previously described method [53]. Intestine myeloperoxidase (MPO) content was determined by procedures suggested by Sahoo et al. [54]. The activity of antioxidant enzymes including catalase (CAT), glutathione reductase (GR), glutathione peroxidase (GPx), and superoxide dismutase (SOD) was measured in liver tissue using commercially available kits (ZellBio GmbH, Hinter den Gärten 56, Lonsee, Germany). Malondialdehyde (MDA) content was assayed using the corresponding kits (ZellBio GmbH, Lonsee, Germany) according to the manufacturer’s protocol. Skin mucus alkaline phosphatase activity (ALP) and total protein level were assayed by the procedures described by Hoseinifar et al. [4] and using commercial kits (Pars Azmun Co., Karaj, Iran). The study of mucosal protease activity was performed following the method described by Ross et al. [55].

### 2.7. Statistical Analyses

The data normality and homogeneity of variance were checked by Shapiro-Wilk and Levene’s tests, respectively. Differences among the experimental groups were assessed by one-way analysis of variance (ANOVA) followed by Duncan’s multiple range test at the level of *p* < 0.05 and presented as means ± standard error (SE). The statistical analysis was performed using SPSS software ver. 19 (SPSS Inc., Chicago, IL, USA).

## 3. Results

The growth performance of fish fed with different levels of synbiotic Biomin^®^IMBO and/or Persian shallot is presented in Table 1. All treatments significantly showed higher FW, WG, WG (%), and SGR compared to the fish fed on the control diet. The maximum FW was recorded in T6 and T7 (*p* < 0.05), whereas the minimum value was in the control group. In addition, the highest and the lowest WG, WG (%), and SGR values were observed in T7 and control groups, respectively. Moreover, the lowest FCR was noticed in fish fed on a diet supplemented with 1% Persian shallot and synbiotic (T6) followed by 3% Persian shallot and 3% synbiotic (T7), whereas the fish fed with control and 1% synbiotic (T2) diets showed the highest FCR level. No mortality was observed in fish of different treatments during the rearing period (*p* > 0.05).

Diets supplemented with different levels of synbiotic Biomin^®^IMBO and/or Persian shallot could significantly improve intestine immune parameters including lysozyme, ACH50, total Ig, and MPO of zebrafish compared to fish fed on the control diet (*p* < 0.05; Table 2).

The results of liver antioxidant parameters of zebrafish fed on diets with different levels of synbiotic Biomin^®^IMBO and/or Persian shallot are presented in Table 3. In all experimental groups the hepatic CAT, SOD, GPx, and GR activities significantly increased compared to the control group. Whereas the highest MDA level was observed in the control group compared to the other treatments (*p* < 0.05). 

Diets supplemented with different levels of synbiotic Biomin^®^IMBO and/or Persian shallot significantly improved skin mucus immune parameters of zebrafish compared to fish fed on the control diet (*p* < 0.05; Figure 1). So that the highest lysozyme values were observed in T4 followed by T5, whereas the lowest value was recorded in the control group. Moreover, the highest and lowest total Ig levels were registered in T7 and T1, respectively. The T2 and T5 groups showed the highest and the control group showed the lowest skin mucus total protein levels. The maximum ALP levels were recorded in T7, T6, and T5, compared to the other groups (*p* < 0.05). Finally, the highest protease values were observed in T3 followed by T2, and its lowest value was recorded in the control group.

## 4. Discussion

The aquaculture industry has been facing serious challenges during the past decade, due to the emergence of resistant pathogens and the limitations in the use of chemicals for disease treatment [56]. Several studies have demonstrated that dietary supplements such as probiotics, prebiotics, and synbiotics provide nonspecific disease protection and act as growth-promoting factors [57]. In cases where probiotics, prebiotics, and synbiotics have increased the survival of aquatic animals, the survival of animals was highest in the probiotic treatment, followed by the prebiotic and synbiotic ones [58]. 

In this study, the results clearly show that the synbiotic Biomin^®^IMBO and/or Persian shallot had beneficial effects on the growth parameters of zebrafish, with higher final FW, WG, WG (%), and SGR compared with the fish fed on the control diet. The best FCR values were observed in fish fed with 1% synbiotic supplemented diet. These outcomes are in agreement with Mehrabi et al. [59], who reported that rainbow trout supplemented with synbiotic showed a significant increase in FW, WG (%), and SGR compared to the control group. The enhanced growth performance may be due to the increased digestive enzyme activity induced by the synbiotic [60]. Probiotic and prebiotic supplements improve the productivity of food through various mechanisms and consequently increases fish growth. One possible mechanism can reside in the fact that prebiotics and probiotics improve the microbial flora of the host gastrointestinal tract towards fermenting bacteria [61].

Plant extracts have been shown to increase digestibility and nutrient availability, leading to increased protein synthesis [62]. Mooseer extract improves the digestion and absorption of dietary proteins by acting on liver enzymes, especially proteases [63].

Sheikhzadeh and Heidarieh [21] declared that synbiotic Biomin^®^IMBO could improve the growth performance and skin mucus immunity in *Carassius carassius*. Firouzbakhsh et al. [64] reported in rainbow trout (*Oncorhynchus mykiss*) significant increases in FW and SGR at all experimental treatments. In their study, the best FCR, and maximum SR were observed in fish fed with 1% synbiotic supplemented diet. Akbary et al. [65] have found that garlic extract in the diet increased final length and weight, SGR, protein efficiency ratio, and protein production ratio in *Mugil cephalus*. Also, the lowest fat and moisture levels and the highest protein and ash levels in body composition were observed in *Mugil cephalus* fed the diet containing 3% garlic extract. 

Additionally, it can be assumed that after feeding the fish with Biomin^®^IMBO supplementation, the bacteria in synbiotics have succeeded in competing with the existing microflora, so that effective colonies were formed in their intestine. Therefore, it is possible that, due to the efficiency of the gastrointestinal tract, the digestion and absorption of diets increased, leading to the improvement of growth and nutrition indices in the fish [66].

In this study, the synbiotic-fed zebrafish showed higher SR in all treatments compared with the control. This finding is in agreement with the results reported by Merrifield et al. [67] and Mehrabi et al. [58] where they showed diets supplemented with symbiotic significantly improved the SR of rainbow trout and salmonids, respectively. Mahghani et al. [68] found that different synbiotic levels of Biomin^®^IMBO could significantly improve growth performance and nutritional indices of *Carassius auratus gibelio*. One possible mechanism is that the consumed probiotics (via synbiotic), during the fermentation process, produce metabolites (i.e., some vitamins, acetate, and lactate) that ultimately improve liver function and the secretion of digestive enzymes, which lead to better absorption of nutrients and growth of fish [69]. Studies have shown that the activity of digestive enzymes affect on the stage of life, chemical composition of the feed and the nutritional needs of fish [40].

Diets with different levels of synbiotic Biomin^®^IMBO supplemented and/or Persian shallot could significantly improve gut immune parameters including lysozyme, ACH50, total Ig, and MPO of zebrafish. Previous studies have shown that the use of synbiotics in the diet leads to an increase in the activity of serum lysozyme and mucus [70]. It is well documented that the mucosal immune system of fish could be boosted by dietary administration of prebiotics, probiotics, and medicinal plants [71]. For instance, Safari et al. [72] evaluated the effect of *Coriandrum sativum* L. on mucosal immune parameters, antioxidant defense in zebrafish and found that it can be proposed that dietary coriander powder can improve mucosal immune parameters and immune and antioxidant genes expression. Also, Karimi Pashaki et al. [73] found that the dietary garlic extract had a positive effect on survival rate, blood and immune parameters changes, and disease resistance of common carp.

Immunoglobulins are natural antibodies that are produced in the absence of external antigenic stimuli, and this feature has become an important part of the non-specific immune system of fish [74]. Therefore, elevation of total Ig could be due to the improvement of immune parameters in response to the addition of synbiotic and/or Persian shallot to zebrafish diet [75].

Jahanjoo et al. [76] showed that serum lysozyme activity, total Ig, and ACH50 activity significantly increased in fish fed a mixture of three medicinal herbs and ginger. The immune system is stimulated by synbiotic Biomin^®^IMBO and increases the proliferation of macrophages, growth, survival of fish, and the volume of beneficial bacteria in the gut, affecting the composition of the body [77]. Proteases are a group of enzymes with the catalytic property of hydrolyzing peptide bonds. Different types of proteases are known in fish mucus that play an important role in the innate immune system of fish [78]. Consistent with the results of this study, previous studies have shown that the use of synbiotics in the diet increases serum lysozyme and mucus in fish [72].

The fish skin mucus provides the first line of defense against pathogens. Several herbal preparations have been reported to enhance immune responses in fish [79]. Some studies reported that skin mucosal immunity is related to dietary Myrtle administration [80]. For instance, the results of Imanpour et al. [81] study showed that the use of ginger increased the mucosal parameters of *Rutilus kutum* and skin mucus immune parameters in the dietary treatments also revealed the same results for lysozyme, protease activity, and total Ig. Likewise, it has been reported that supplementation of the fish diet with numerous plants could increase skin mucus immune responses [50,82]. In the study of Soleimani et al. [83], the results showed that garlic powder increased the amount of alkaline phosphatase and soluble protein levels of skin mucus. Increased levels of mucus and serum proteins are good indicators for assessing the nonspecific immune status of fish. An increment in the activity of skin mucus ALP has been reported in roach (*Rutilus rutilus*) fries fed garlic-supplemented diet [84]. Although the ability of garlic to stimulate benefits in fish has been demonstrated [85], to our knowledge, there are no studies on complement activity evaluation in fish fed with Persian shallot.

SOD, CAT, and GPx are important antioxidant enzymes, which play vital roles in counteracting the toxicity of reactive oxygen species (ROS) under unfavorable conditions [86]. According to our results, all experimental groups showed a significant increase in the hepatic CAT, SOD, GPx, and GR activities in comparison to the control group. Whereas the highest MDA level was observed in the control group compared to the other treatments. In fact, these protective systems (antioxidant enzymes and oxidative stress) are able to balance the production and removal of ROS under natural conditions. Disruption of this process leads to disruption of the body’s homeostasis system and causes oxidative stress in various cells of living organisms [87]. SOD, CAT, and GPx enzymes involved in the antioxidant system are identified as the first defense mechanism against oxidative stress through radical processes and phagocytosis within damaged tissue [88]. However, the major role of these enzymes is to neutralize the ROS produced in different organs [89]. Jafarinejad et al. [90] investigated the effect of ginger on zebrafish and found that SOD, CAT, and GPx enzyme activities were significantly higher in the groups fed 2 and 5% ginger diets.

When two supplements are used together, they may act as an additive synergism or potentiation. The additive effect occurs when the effect of two substances used together is measured by the sum of the individual effects. The synergistic effect occurs when the combined effect of the two products is significantly greater than the sum of the effects of each factor alone. The use of the term has different potentials; some pharmacologists use it to describe more of the synergistic effect than the cumulative effect, and some use the term to describe the effect of two compounds being present simultaneously. In addition to changing the intestinal bacterial balance towards beneficial bacteria, these substances have very favorable effects on the intestinal tissue and liver of fish [16].

## 5. Conclusions

According to the results of the present study, it can be concluded that the supplement of synbiotic Biomin^®^IMBO and/or Persian shallot in the diet of zebrafish can be effective for increasing the growth performances and survival rate and improving the immune responses of zebrafish. Further studies are recommended to investigate the effects of these synbiotics on the health and survival of fish exposed to environmental stress and infectious diseases.

## Figures and Tables

**Figure 1 animals-11-02995-f001:**
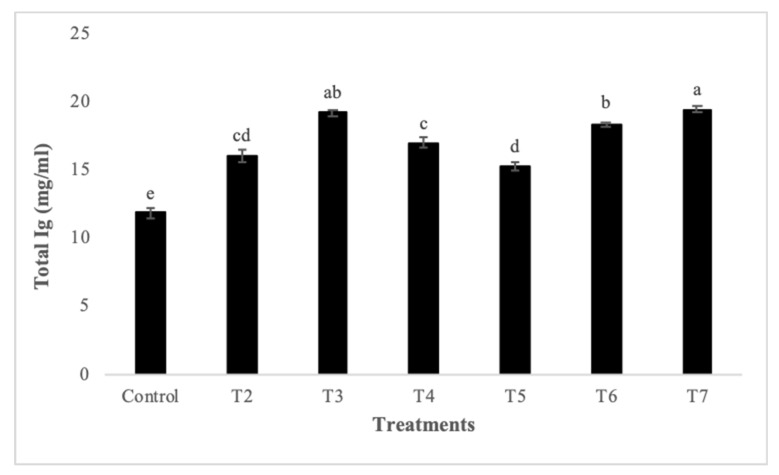
Skin mucus immune parameters of zebrafish fed on diets with different levels of synbiotic Biomin^®^IMBO and/or Persian shallot for 60 days. T1, control (without any supplementation); T2, 1% synbiotic; T3, 3% synbiotic; T4, 1% Persian shallot; T5, 3% Persian shallot; T6, 1% Persian shallot and 1% synbiotic; T7, 3% Persian shallot and 3% synbiotic. Data are expressed as the mean ± SE. Different letters (a–e) in the same row indicate significant differences among the treatments (*p* < 0.05).

**Table 1 animals-11-02995-t001:** Growth performance of zebrafish fed on diets with different levels of synbiotic Biomin^®^IMBO and/or Persian shallot for 60 days. T1, control (without any supplementation); T2, 1% synbiotic; T3, 3% synbiotic; T4, 1% Persian shallot; T5, 3% Persian shallot; T6, 1% Persian shallot and 1% synbiotic; T7, 3% Persian shallot and 3% synbiotic.

Parameters	Control	T2	T3	T4	T5	T6	T7
IW (mg)	151.90 ± 0.83	151.95 ± 0.61	152.51 ± 0.69	151.50 ± 0.98	152.09 ± 1.30	152.06 ± 1.04	151.33 ± 1.11
FW (mg)	301.68 ± 2.34 ^d^	309.76 ± 1.67 ^c^	314.42 ± 1.11 ^ab^	310.42 ± 0.48 ^bc^	315.96 ± 1.09 ^a^	314.08 ± 1.12 ^a–c^	317.68 ± 0.89 ^a^
WG (mg)	149.77 ± 2.52 ^d^	157.81 ± 1.09 ^c^	161.90 ± 0.48 ^a–c^	158.92 ± 1.04 ^bc^	163.87 ± 1.62 ^ab^	162.02 ± 1.60 ^a–c^	166.35 ± 2.00 ^a^
WG (%)	98.60 ± 1.91 ^c^	103.85 ± 0.35 ^b^	106.15 ± 0.30 ^ab^	104.91 ± 1.31 ^ab^	107.77 ± 1.85 ^ab^	106.56 ± 1.65 ^ab^	109.95 ± 2.13 ^a^
SGR (% d^−1^)	1.14 ± 0.016 ^c^	1.18 ± 0.002 ^b^	1.20 ± 0.002 ^ab^	1.19 ± 0.01 ^ab^	1.21 ± 0.014 ^ab^	1.20 ± 0.013 ^ab^	1.23 ± 0.016 ^a^
FCR	1.91 ± 0.006 ^a^	1.89 ± 0.008 ^a^	1.77 ± 0.008 ^c^	1.81 ± 0.01 ^b^	1.75 ± 0.006 ^cd^	1.72 ± 0.008 ^e^	1.74 ± 0.005 ^de^
SR (%)	100 ± 0.00	100 ± 0.00	100 ± 0.00	100 ± 0.00	100 ± 0.00	100 ± 0.00	100 ± 0.00

IW: initial weight; FW: final weight; WG: weight gain; SGR: specific growth rate; FCR: feed conversion ratio; SR: survival rate. Data are expressed as the mean ± SE. Different letters (a–e) in the same row indicate significant differences among the treatments (*p* < 0.05).

**Table 2 animals-11-02995-t002:** Intestine immune parameters of zebrafish fed on diets with different levels of synbiotic Biomin^®^IMBO and/or Persian shallot for 60 days. T1, control (without any supplementation); T2, 1% synbiotic; T3, 3% synbiotic; T4, 1% Persian shallot; T5, 3% Persian shallot; T6, 1% Persian shallot and 1% synbiotic; T7, 3% Persian shallot and 3% synbiotic.

Parameters	T1	T2	T3	T4	T5	T6	T7
Lysozyme (U/mg prot)	12.77 ± 0.24 ^d^	16.08 ± 0.15 ^c^	18.93 ± 0.29 ^b^	16.99 ± 0.34 ^c^	21.88 ± 0.39 ^a^	18.43 ± 0.49 ^b^	20.96 ± 0.89 ^a^
ACH50 (U/mg prot)	35.80 ± 0.91 ^d^	45.18 ± 1.31 ^bc^	41.97 ± 1.30 ^c^	47.86 ± 1.00 ^ab^	49.80 ± 1.12 ^a^	45.21 ± 0.72 ^bc^	50.88 ± 0.76 ^a^
Total Ig (mg/mL)	33.21 ± 0.78 ^e^	42.56 ± 1.32 ^bc^	44.62 ± 1.21 ^b^	40.73 ± 0.79 ^c^	37.24 ± 0.94 ^d^	47.78 ± 0.72 ^a^	50.58 ± 1.25 ^a^
MPO (OD at 450 nm)	1.31 ± 0.12 ^c^	2.81 ± 0.08 ^a^	2.31 ± 0.18 ^b^	2.95 ± 0.09 ^a^	3.12 ± 0.07 ^a^	2.79 ± 0.07 ^a^	2.92 ± 0.04 ^a^

Data are expressed as the mean ± SE. Different letters (a–e) in the same row indicate significant differences among the treatments (*p* < 0.05).

**Table 3 animals-11-02995-t003:** Liver antioxidant parameters of zebrafish fed on diets with different levels of synbiotic Biomin^®^IMBO and/or Persian shallot for 60 days. T1, control (without any supplementation); T2, 1% synbiotic; T3, 3% synbiotic; T4, 1% Persian shallot; T5, 3% Persian shallot; T6, 1% Persian shallot and 1% synbiotic; T7, 3% Persian shallot and 3% synbiotic.

Parameters	Control	T2	T3	T4	T5	T6	T7
CAT (U/mg prot)	62.19 ± 1.31 ^c^	71.99 ± 1.34 ^ab^	74.04 ± 2.03 ^a^	73.54 ± 1.53 ^a^	67.62 ± 1.00 ^b^	70.18 ± 1.50 ^ab^	67.74 ± 1.36 ^b^
SOD (U/ mg prot)	74.88 ± 1.51 ^d^	86.39 ± 1.05 ^ab^	81.37 ± 1.03 ^c^	82.76 ± 1.33 ^bc^	89.99 ± 1.44 ^a^	81.21 ± 1.09 ^c^	82.32 ± 1.22 ^c^
MDA (nmol/ mg prot)	3.37 ± 0.13 ^a^	2.85 ± 0.03 ^b^	2.65 ± 0.12 ^b^	2.53 ± 0.13 ^bc^	2.62 ± 0.09 ^b^	2.28 ± 0.08 ^cd^	2.11 ± 0.08 ^d^
GPx (U/ mg prot)	42.54 ± 0.95 ^e^	60.16 ± 1.14 ^c^	59.56 ± 1.03 ^c^	54.27 ± 1.24 ^d^	62.21 ± 2.05 ^bc^	67.98 ± 0.64 ^a^	64.11 ± 0.75 ^b^
GR (U/ mg prot)	67.62 ± 0.97 ^e^	76.13 ± 1.19 ^d^	80.59 ± 0.72 ^bc^	87.88 ± 1.13 ^a^	77.56 ± 0.81 ^cd^	82.97 ± 1.35 ^b^	76.08 ± 1.23 ^d^

Data are expressed as the mean ± SE. Different letters (a–e) in the same row indicate significant differences among the treatments (*p* < 0.05).

## Data Availability

Data available on request due to restrictions, e.g., privacy or ethical. The data presented in this study are available on request from the corresponding author. The data are not publicly available due to the law of the Ministry of Science Research and Technology.

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
