# Peer review of "Combined and Singular Effects of Ethanolic Extract of Persian Shallot (Allium hirtifolium Boiss) and Synbiotic Biomin®IMBO on Growth Performance, Serum- and Mucus-Immune Parameters and Antioxidant Defense in Zebrafish (Danio rerio)"

_animals, 2021, doi:10.3390/ani11102995_

Round 1
Reviewer 1 Report
The research was on the effect of Persian shallot (Allium hirtifolium Boiss) and synbiotic Biomin®IMBO on the growth performance and the immune parameters of zebra fish. The design and the methods of the research was appropriate, and it is suitable to be published on Animals. However, there are few questions to be answered.
- The authors used Persian shallot alone or in combine with Biomin® IMBO. Would the authors explain more the reason of using Persian shatllot in diet? Although it is well known for high antioxidants contain, it is a plant mainly distributed in Iran, is the production sufficient for the use in aquaculture? Besides, why the authors wants to combine this plant with Biomin® IMBO?
- One-way ANOVA was performed in this research. However, one-way ANOVA uses only ONE independent variable, but in this research, two independent variable were involved (the level of Persian shallot and the level of Biomin® IMBO). Thus, two-way ANOVA would be more suitable for the analysis.
- Please align the content in the Table 2 and Table 3.
- Line 88, please delete one space between “from” and ”moderate”.
- Line 89, please move “[32]” to the end of the sentence.
Author Response
First, we would like to thanks you for your valuable comments, we have revised according to your points and provide answer for your questions :
- The authors used Persian shallot alone or in combine with Biomin® IMBO. Would the authors explain more the reason of using Persian shatllot in diet?
Response: Persian shallot is a potent antioxidant and synbiotic Biomin®IMBO is a good growth/immune modulator. Thus, we aimed to examine their additive effects on the well-being and overall performance of zebrafish.
- Although it is well known for high antioxidants contain, it is a plant mainly distributed in Iran, is the production sufficient for the use in aquaculture?
Response: Yes, it is native and endemic to Iran and grows as a wild plant across the Zagross Mountains and southern area. Also, Esfahan state is the biggest polar in the cultivation of shallots that the area under cultivation of this crop will reach one hundred and fifty hectares this year. Moreover, it is a self-growth plant that is drought-resistant, and it is growable in the majority of areas.
Besides, why the authors want to combine this plant with Biomin® IMBO?
Response: As we mentioned above, Persian shallot is a potent antioxidant and synbiotic Biomin®IMBO is a good growth/immune modulator. Thus, we aimed to examine their additive effects on the well-being and overall performance of zebrafish.
- One-way ANOVA was performed in this research. However, one-way ANOVA uses only ONE independent variable, but in this research, two independent variables were involved (the level of Persian shallot and the level of Biomin® IMBO). Thus, two-way ANOVA would be more suitable for the analysis.
Response: many thanks for your valuable feedback, indeed considering our study design was based on the additive effects (not synergistic effects) so we don’t have sufficient treatments to run a two-way ANOVA analysis. In fact, due to limitations, we couldn’t consider more treatments. It's worthy to mention that, two-way can be applied when there is the assumption that there should be Interaction Effect between two independent variables. Given the fact that our assumption was that there is no Interaction Effect between synbiotic and Persian shallot, we designed the study as such and used one-way ANOVA
Please align the content in the Table 2 and Table 3.
Response: They were checked and revised.
- Line 88, please delete one space between “from” and ”moderate”.
Response: They were checked and revised.
- Line 89, please move “[32]” to the end of the sentence.
Response: They were checked and revised.
Reviewer 2 Report
- Since single additives can improve the growth and immune status of zebrafish, what is the significance of using compound additives?
- Zebrafish can be used as a model animal, but it does not seem to represent aquatic economic animals.
- Is there any reference for selecting the concentration of additives?
- Were the same particle size feeds fed during the 60-day period? As we all known, different growth stages of fish should use different particle size of feed.
- Line 74: “zebrafish”
- Line 85-87: The study was published in 2012, so it isn’t proper to be called as “recent study”.
- Line 89: Why there was “activity” at the end of the sentence?
- There are extra spaces between words, like “from moderate” (line 88), “have positive” (line 90), “from solid” (line 113).
- Why was the formula of “Survival rate” in different font?
- Line 295: Why do not use “ROS” replace with “reactive oxygen species”, since you have used “ROS” in line 290?
- Line 366: Why is there no reference 17?
- There are too many references.
- Table 2/3: The list is a little confusing. Why are the numbers in each row not on the same horizontal line?
Author Response
we would like to thanks you for your valuable feedback, we have revised the ms according to your comments, our response to your questions can be found below
- Since single additives can improve the growth and immune status of zebrafish, what is the significance of using compound additives?
Response: Persian shallot is a potent antioxidant and synbiotic Biomin®IMBO is a good growth/immune modulator. Thus, we aimed to examine the additive effects of these feed additives on the well-being and overall performance of zebrafish.
- Zebrafish can be used as a model animal, but it does not seem to represent aquatic economic animals.
Response: Over the past decades, the zebrafish (Danio rerio) has attracted considerable attention as an excellent zebrafish as an animal model for biomedical research. They are used to study genetic evolution and, more recently, to understand human diseases and to screen for therapeutic drugs. In addition, working with zebrafish for research is easier than farmed fish and can be used for preliminary experiments. And if positive results are obtained from preliminary experiments, further studies can be designed and performed on farmed fish and/or more evolved animals. Also, the species is one of the attractive and well-known ornamental fish. With special importance and is an economical fish for ornamental fish aquaculture.
- Is there any reference for selecting the concentration of additives?
Response: Yes, there are several references as follows:
* Sheikhzadeh, N, & Heidarieh, M. Effects of commercial synbiotic Biomin®IMBO on growth performance, some biochemical parameters and skin mucosal immunity in crucian carp (Carassius carassius). Journal of Aquaculture Development, 2020, 14 (2), 55-65.
* Jahari, M.A., Mustafa, S., Roslan, M.A., Manap, Y.A., Lamasudin, D., & Jamaludin, F.I. The Effects of Synbiotics and Probiotics Supplementation on Growth Performance of Red Hybrid Tilapia, Oreochromis mossambicus x Oreochromis niloticus. Journal of Biochemistry, Microbiology and Biotechnology, 2018, 6(1), 5-9.
* Sewaka, M., Trullas, C., Chotiko, A., Rodkhum, C., Chansue, N., Boonanuntanasarn, S., Pirarat, N. Efficacy of symbiotic Jerusalem artichoke and Lactobacillus rhamnosus GG supplemented diets on growth performance, serum biochemical parameters, intestinal morphology, immune parameters and protection against Aeromonas veronii in juvenile red tilapia (Oreochromis spp.). Fish and Shellfish Immunology, 2019, 86, 260–268.
* Changizi, R., Manouchehri, H., Hosseinifard, M., & Ghiasvand, Z. Effect of Different Levels of Biomin Imbo Synbiotic on Growth Indices, Feeding Factors and Survival Rate of Green Terror (Andinoacara rivulatus). Journal of Animal Environment, 2019, 11(3), 135-140.
* Gheshlaghi, P., Rashidian, G., Chardeh Baladehi, E., Bagheri, T., & Ghafari Farsani, H. Effect of synbiotic biomin imbo on growth parameters, survival, digestive enzymes and mucus parameters of banded cichlide (Heros severus). Aquatic Physiology and Biotechnology, 2015, 3(1), 49-74.
Were the same particle size feeds fed during the 60-day period? As we all know, different growth stages of fish should use different particle sizes of feed.
Response: In order to ensure the quality of the experimental diets (basal diet supplemented with corresponded feed additives) made for the fish in this study, experimental diets were made every 20 days. At each stage, according to the growth of fish, the appropriate size of the commercial diet was used to prepare experimental diets. It has been added to the text (lines: 144-147).
- Line 74: “zebrafish”
Response: They were checked and revised.
- Line 85-87: The study was published in 2012, so it isn’t proper to be called as “recent study”.
Response: They were checked and revised.
- Line 89: Why there was “activity” at the end of the sentence?
Response: They were checked and revised.
- There are extra spaces between words, like “from moderate” (line 88), “have positive” (line 90), “from solid” (line 113).
Response: They were checked and revised.
- Why was the formula of “Survival rate” in a different font?
Response: They were checked and revised.
- Line 295: Why do not use “ROS” replace with “reactive oxygen species”, since you have used “ROS” in line 290?
Response: They were checked and revised.
- Line 366: Why is there no reference 17?
Response: They were checked and revised.
- There are too many references.
Response: We tried to remove the extra references as much as possible.
- Table 2/3: The list is a little confusing. Why are the numbers in each row not on the same horizontal line?
Response: They were checked and revised.
Reviewer 3 Report
I found a lot of minor, but mainly editorial mistakes. I marked they in the text as comments.

Author Response
we would like to thanks you for your valuable feedback, we have revised the ms according to your comments, our response to your questions can be found below
I found a lot of minor, but mainly editorial mistakes. I marked they in the text as comments.
- Add simple summary
Response: Now we added it.
- Small letter
Response: They were checked and revised.
- Change font
Response: They were checked and revised.
- One paragraph should be added. The authors should highly present that combination of different additives an improve health, stress responce or e.g. reproduction parameters - use also e.g. DOI: 10.1089/zeb.2017.1416
Response: They were checked and revised.
- It should be in one line
Response: They were checked and revised.
- Reformat table
Response: They were checked and revised.
Reviewer 4 Report
This manuscript nicely described about the use of synbiotics and natural products on improving the immune status of Zebrafish. The results clearly indicates the positive role of these compounds on enhanced immunity. This manuscript maybe accepted in the current form. But, it would have been an excellent findings, if the authored could have performed disease challenge study in zebrafish to confirm the health beneficial role of these products on zebrafish growth performance, disease resistance and immunity.
Author Response
we would like to thanks your valuable comment and we are happy to see that you find our study of interest. due to some limitations, it was not possible for us to perform disease challenges in this project. we will consider your valuable feedback in our future projects.
Round 2
Reviewer 2 Report
Good job.